# Thermodynamics of the Ising Model Encoded in Restricted Boltzmann Machines

**DOI:** 10.3390/e24121701

**Published:** 2022-11-22

**Authors:** Jing Gu, Kai Zhang

**Affiliations:** 1Division of Natural and Applied Sciences, Duke Kunshan University, Kunshan 215300, China; 2Data Science Research Center (DSRC), Duke Kunshan University, Kunshan 215300, China

**Keywords:** restricted Boltzmann machine, Ising model, machine learning, statistical physics, phase transition, entropy estimation

## Abstract

The restricted Boltzmann machine (RBM) is a two-layer energy-based model that uses its hidden–visible connections to learn the underlying distribution of visible units, whose interactions are often complicated by high-order correlations. Previous studies on the Ising model of small system sizes have shown that RBMs are able to accurately learn the Boltzmann distribution and reconstruct thermal quantities at temperatures away from the critical point $T_{c}$. How the RBM encodes the Boltzmann distribution and captures the phase transition are, however, not well explained. In this work, we perform RBM learning of the 2d and 3d Ising model and carefully examine how the RBM extracts useful probabilistic and physical information from Ising configurations. We find several indicators derived from the weight matrix that could characterize the Ising phase transition. We verify that the hidden encoding of a visible state tends to have an equal number of positive and negative units, whose sequence is randomly assigned during training and can be inferred by analyzing the weight matrix. We also explore the physical meaning of the visible energy and loss function (pseudo-likelihood) of the RBM and show that they could be harnessed to predict the critical point or estimate physical quantities such as entropy.

## 1. Introduction

The tremendous success of deep learning in multiple areas over the last decade has really revived the interplay between physics and machine learning, in particular neural networks [1]. On the one hand, (statistical) physics ideas [2], such as the renormalization group (RG) [3], the energy landscape [4], free energy [5], glassy dynamics [6], jamming [7], Langevin dynamics [8], and field theory [9], shed some light on the interpretation of deep learning and statistical inference in general [10]. On the other hand, machine learning and deep learning tools are harnessed to solved a wide range of physics problems, such as interaction potential construction [11], phase transition detection [12,13], structure encoding [14], physical concepts’ discovery [15], and many others [16,17]. At the very intersection of these two fields lies the restricted Boltzmann machine (RBM) [18], which serves as a classical paradigm to investigate how an overarching perspective could benefit both sides.

The RBM uses hidden–visible connections to encode (high-order) correlations between visible units [19]. Its precursor—the (unrestricted) Boltzmann machine—was inspired by spin glasses [20,21] and is often used in the inverse Ising problem to infer physical parameters [22,23,24]. The restriction of hidden–hidden and visible–visible connections in RBMs allows for more efficient training algorithms and, therefore, leads to recent applications in Monte Carlo simulation acceleration [25], quantum wavefunction representation [26,27], and polymer configuration generation [28]. Deep neural networks formed by stacks of RBMs have been mapped onto the variational RG due to their conceptual similarity [29]. RBMs are also shown to be equivalent to tensor network states from quantum many-body physics [30] and interpretable in light of statistical thermodynamics [31,32,33]. As simple as it seems, energy-based models like the RBM could eventually become the building blocks of autonomous machine intelligence [34].

Besides the above-mentioned efforts, the RBM has also been applied extensively in the study of the minimal model for second-order phase transition—the Ising model. For the small systems under investigation, it was found that RBMs with an enough hidden units can encode the Boltzmann distribution, reconstruct thermal quantities, and generate new Ising configurations fairly well [35,36,37]. The visible → hidden → visible ⋯ generating sequence of the RBM can be mapped onto an RG flow in physical temperature (often towards the critical point) [38,39,40,41,42]. However, the mechanism and power of the RBM to capture physics concepts and principles have not been fully explored. First, in what way is the Boltzmann distribution of the Ising model learned by the RBM? Second, can the RBM learn and even quantitatively predict the phase transition without extra human knowledge? An affirmative answer to the second question is particularly appealing, because simple unsupervised learning methods such as principal component analysis (PCA) using configuration information alone do not provide quantitative prediction for the transition temperature [43,44,45] and supervised learning with neural networks requires human labeling of the phase type or temperature of a given configuration [46,47].

In this article, we report a detailed numerical study on RBM learning of the Ising model with a system size much larger than those used previously. The purpose is to thoroughly dissect the various parts of the RBM and reveal how each part contributes to the learning of the Boltzmann distribution of the input Ising configurations. Such understanding allows us to extract several useful machine learning estimators or predictors for physical quantities, such as entropy and phase transition temperature. Conversely, the analysis of a physical model helps us to obtain important insights about the meaning of RBM parameters and functions, such as the weight matrix, visible energy, and pseudo-likelihood. Below, we first introduce our Ising datasets and the RBM and its training protocols in Section 2. We then report and discuss the results of the model parameters, hidden layers, visible energy, and pseudo-likelihood in Section 3. After the conclusion, more details about the Ising model and the RBM are provided in the Appendix B, Appendix C and Appendix D. Sample codes of the RBM are shared on GitHub at https://github.com/Jing-DS/isingrbm (accessed on 18 November 2022).

## 2. Models and Methods

### 2.1. Dataset of Ising Configurations Generated by Monte Carlo Simulations

The Hamiltonian of the Ising model with $N=L^{d}$ spins in a configuration $s={\left[s_{1},s_{2},\cdots ,s_{N}\right]}^{T}$ on a *d*-dimensional hypercubic lattice of linear dimension *L* in the absence of a magnetic field is
(1)$$\begin{array}{c} H\left(s\right)=-J\sum_{\langle i,j\rangle }s_{i}s_{j} \end{array}$$
where the spin variable $s_{i}=\pm 1$ (i=1,2,⋯,N), the coupling parameter J>0 (set to unity) favors ferromagnetic configurations (parallel spins), and the notation 〈i,j〉 means to sum over nearest neighbors [48]. At a given temperature *T*, the configuration s drawn from the sample space of $2^{N}$ states follows the Boltzmann distribution
(2)$$\begin{array}{c} p_{T}\left(s\right)=\frac{e^{-\frac{H(s)}{k_{B}T}}}{Z_{T}} \end{array}$$
where $Z_{T}=\sum_{s}e^{-\frac{H(s)}{k_{B}T}}$ is the partition function. The Boltzmann constant $k_{B}$ is set to unity.

Using single-flip Monte Carlo simulations under periodic boundary conditions [49], we generate Ising configurations for two-dimensional (2d) systems (d=2) of L=64 (N=4096) at $n_{T}=16$ temperatures T=0.25,0.5,0.75,1.0,⋯,4.0 (in units of $J/k_{B}$) and for three-dimensional (3d) systems (d=3) of L=16 (N=4096) at $n_{T}=20$ temperatures T=2.5,2.75, 3.0,3.25, 3.5,3.75,4.0, 4.25,4.3,4.4,4.5, 4.6,4.7,4.75,5.0,5.25, 5.5,5.75,6.0,6.25. After being fully equilibrated, M= 50,000 configurations at each *T* are collected into a dataset $D_{T}$ for that *T*. For 2d systems, we also use a dataset $D_{\cup T}$ consisting of 50,000 configurations per temperature from *all T*s.

Analytical results of the thermal quantities of the 2d Ising model, such as internal energy 〈E〉, (physical) entropy *S*, heat capacity $C_{V}$, and magnetization 〈m〉, are well known [50,51,52,53]. Numerical simulation methods and results of the 3d Ising model have also been reported [54]. The thermodynamic definitions and relations used in this work are summarized in Appendix B.

### 2.2. Restricted Boltzmann Machine

The restricted Boltzmann machine (RBM) is a two-layer energy-based model with $n_{h}$ hidden units (or neurons) $h_{i}=\pm 1$ ($i=1,2,\cdots ,n_{h}$) in the hidden layer, whose state vector is $h={\left[h_{1},h_{2},\cdots ,h_{n_{h}}\right]}^{T}$, and $n_{v}$ visible units $v_{j}=\pm 1$ ($j=1,2,\cdots ,n_{v}$) in the visible layer, whose state vector is $v={\left[v_{1},v_{2},\cdots ,v_{n_{v}}\right]}^{T}$ (Figure 1) [55]. In this work, the visible layer is just the Ising configuration vector, i.e., v=s, with $n_{v}=N$. We chose the binary unit {−1,+1} (instead of {0,1}) to better align with the definition of Ising spin variable $s_{i}$.

The total energy $E_{\theta }\left(v,h\right)$ of the RBM is defined as
(3)$$\begin{array}{cc} E_{\theta }\left(v,h\right) & =-b^{T}v-c^{T}h-h^{T}Wv\\ \; & =-\sum_{j=1}^{n_{v}}b_{j}v_{j}-\sum_{i=1}^{n_{h}}c_{i}h_{i}-\sum_{i=1}^{n_{h}}\sum_{j=1}^{n_{v}}W_{ij}h_{i}v_{j} \end{array}$$
where $b={\left[b_{1},b_{2},\cdots ,b_{n_{v}}\right]}^{T}$ is the visible bias, $c={\left[c_{1},c_{2},\cdots ,c_{n_{h}}\right]}^{T}$ is the hidden bias, and
(4)$$W_{n_{h}\times n_{v}}=\left[\begin{array}{c} -w_{1}^{T}-\\ -w_{2}^{T}-\\ \vdots \\ -w_{n_{h}}^{T}- \end{array}\right]=\left[\begin{array}{cccc} | & | & \; & |\\ w_{:,1} & w_{:,2} & \cdots & w_{:,n_{v}}\\ | & | & \; & | \end{array}\right]$$
is the interaction weight matrix between visible and hidden units. Under this notation, each row vector $w_{i}^{T}$ (of dimension $n_{v}$) is a *filter* mapping from the visible state v to a hidden unit *i*, and each column vector $w_{:,j}$ (of dimension $n_{h}$) is an *inverse filter* mapping from the hidden state h to a visible unit *j*. All parameters are collectively written as θ={W,b,c}. “Restricted” refers to the lack of interaction between hidden units or between visible units.

The joint distribution for an overall state (v,h) is
(5)$$p_{\theta }\left(v,h\right)=\frac{e^{-E_{\theta }\left(v,h\right)}}{Z_{\theta }}$$
where the partition function of the RBM:(6)$$Z_{\theta }=\sum_{v}\sum_{h}e^{-E_{\theta }\left(v,h\right)}.$$The learned *model distribution* for visible state v is from the marginalization of $p_{\theta }\left(v,h\right)$:(7)$$p_{\theta }\left(v\right)=\sum_{h}p_{\theta }\left(v,h\right)=\frac{1}{Z_{\theta }}e^{-E_{\theta }\left(v\right)},$$
where the *visible energy* (an effective energy for visible state v (often termed as “free energy” in the machine learning literature)):(8)$$\begin{array}{cc} E_{\theta }\left(v\right)\; & =-b^{T}v-\sum_{i=1}^{n_{h}}\ln\left(e^{-w_{i}^{T}v-c_{i}}+e^{w_{i}^{T}v+c_{i}}\right) \end{array}$$
is defined according to $e^{-E_{\theta }\left(v\right)}=\sum_{h}e^{-E_{\theta }\left(v,h\right)}$ such that $Z_{\theta }=\sum_{v}e^{-E_{\theta }\left(v\right)}.$ See Appendix C for a detailed derivation.

The conditional distributions to generate h from v, $p_{\theta }\left(h|v\right)$, and to generate v from h, $p_{\theta }\left(v|h\right)$, satisfying $p_{\theta }\left(v,h\right)=p_{\theta }\left(h|v\right)p_{\theta }\left(v\right)=p_{\theta }\left(v|h\right)p_{\theta }\left(h\right)$, can be written as products:(9)$$\begin{array}{cc} p_{\theta }\left(h|v\right) & =\prod_{i=1}^{n_{h}}p_{\theta }\left(h_{i}|v\right)\\ p_{\theta }\left(v|h\right) & =\prod_{j=1}^{n_{v}}p_{\theta }\left(v_{j}|h\right) \end{array}$$
because $h_{i}$ are independent of each other (at fixed v) and $v_{j}$ are independent of each other (at fixed h). It can be shown that
(10)$$\begin{array}{cc} p_{\theta }\left(h_{i}=1|v\right) & =\sigma \left(2(c_{i}+w_{i}^{T}v)\right)\\ p_{\theta }\left(h_{i}=-1|v\right) & =1-\sigma \left(2(c_{i}+w_{i}^{T}v)\right)\\ p_{\theta }\left(v_{j}=1|h\right) & =\sigma \left(2(b_{j}+h^{T}w_{:,j})\right)\\ p_{\theta }\left(v_{j}=-1|h\right) & =1-\sigma \left(2(b_{j}+h^{T}w_{:,j})\right) \end{array}$$
where the sigmoid function $\sigma \left(z\right)=\frac{1}{1+e^{-z}}$ (Appendix C).

### 2.3. Loss Function and Training of RBMs

Given the dataset $D={\left[v_{1},v_{2},\cdots ,v_{M}\right]}^{T}$ of *M* samples generated independently from the identical *data distribution*
$p_{D}\left(v\right)$ ($v\overset{i.i.d.}{∼}p_{D}\left(v\right)$), the goal of RBM learning is to find a model distribution $p_{\theta }\left(v\right)$ that approximates $p_{D}\left(v\right)$. In the context of this work, the data samples vs are Ising configurations, and the data distribution $p_{D}\left(v\right)$ is or is related to the Ising–Boltzmann distribution $p_{T}\left(s\right)$.

Based on maximum likelihood estimation, the optimal parameters $\theta^{*}=\arg\min_{\theta }L\left(\theta \right)$ can be found by minimizing the negative log likelihood:(11)$$L\left(\theta \right)={\langle -\ln p_{\theta }\left(v\right)\rangle }_{v∼p_{D}}={\langle E_{\theta }\left(v\right)\rangle }_{v∼p_{D}}+\ln Z_{\theta }$$
which serves as the *loss function* of RBM learning. Note that the partition function $Z_{\theta }$ only depends on the model, not on the data. Since the calculation of $Z_{\theta }$ involves summation over all possible (v,h) states, which is not feasible, L(θ) cannot be evaluated exactly, except for very small systems [56]. Special treatments have to be devised, for example by mean-field theory [57] or by importance sampling methods [58]. An interesting feature of the RBM is that, although the actual loss function L(θ) is not accessible, its gradient:(12)$$\nabla_{\theta }L\left(\theta \right)={\langle \nabla_{\theta }E_{\theta }\left(v\right)\rangle }_{v∼p_{D}}-{\langle \nabla_{\theta }E_{\theta }\left(v\right)\rangle }_{v∼p_{\theta }}$$
can be sampled, which enables a gradient descent learning algorithm. From step *t* to step t+1, the model parameters are updated with learning rate η as
(13)$$\theta_{t+1}=\theta_{t}-\eta \nabla_{\theta }L\left(\theta_{t}\right).$$

To evaluate the loss function, we used its approximate, the pseudo-(negative log) likelihood [59]:(14)$$\tilde{L}\left(\theta \right)={\langle -\sum_{i=1}^{n_{v}}\ln p_{\theta }\left(v_{i}|v_{j\neq i}\right)\rangle }_{v∼p_{D}}\approx L\left(\theta \right)$$
where the notation:(15)$$\begin{array}{c} p_{\theta }\left(v_{i}|v_{j\neq i}\right)=p_{\theta }\left(v_{i}|v_{j}\;\mathrm{for}\;j\neq i\right)=\frac{e^{-E_{\theta }\left(v\right)}}{e^{-E_{\theta }\left(v\right)}+e^{-E_{\theta }\left(\left[v_{1},\cdots ,-v_{i},\cdots ,v_{n_{v}}\right]\right)}} \end{array}$$
is the conditional probability for component $v_{i}$ given that all the other components $v_{j}$(j≠i) are fixed [37]. Practically, to avoid the time-consuming sum over all visible units $\sum_{i=1}^{n_{v}}$, it is suggested to randomly sample one $i_{0}\in \left\{1,2,\cdots ,n_{v}\right\}$ and estimate that:(16)$$\tilde{L}\left(\theta \right)\approx {\langle -n_{v}\ln p_{\theta }\left(v_{i_{0}}|v_{j\neq i_{0}}\right)\rangle }_{v∼p_{D}},$$
if all the visible units are on average translation-invariant [60]. To monitor the reconstruction error, we also calculated the cross-entropy CE between the initial configuration v and the conditional probability $p_{\theta }\left(v^{\prime }|h\right)$ for reconstruction $v\overset{p_{\theta }\left(h|v\right)}{⟶}h\overset{p_{\theta }\left(v^{\prime }|h\right)}{⟶}v^{\prime }$ (see Appendix D for the definition).

For both 2d and 3d Ising systems, we first trained single-temperature RBMs (*T*-RBM). M= 50,000 Ising configurations at each *T* forming a dataset $D_{T}$ are used to train one model such that there are $n_{T}$*T*-RBMs in total. While $n_{v}=N$, we tried various numbers of hidden units with $n_{h}=400,900,1600,2500$ in 2d and $n_{h}=400,900,1600$ in 3d. For 2d systems, we also trained an all-temperature RBM (∪T-RBM) for which 50,000 Ising configurations per temperature are drawn to compose a dataset $D_{\cup T}$ of *M* = 50,000$\;n_{T}=8\times 10^{5}$ samples. The number of hidden units for this ∪T-RBM is $n_{h}=400,900,1600.$ Weight matrix W is initialized with Glorot normal initialization [61] (b and c are initialized as zero). Parameters are optimized with the stochastic gradient descent algorithm of learning rate $\eta =1.0\times 10^{-4}$ and batch size 128. The negative phase (model term) of the gradient ${\langle \nabla_{\theta }E_{\theta }\left(v\right)\rangle }_{v∼p_{\theta }}$ is calculated using CD-k Gibbs sampling with k=5. We stopped the training until $\tilde{L}$ and CE converged, typically at 100–2000 epochs (see the Appendix A). Three Nvidia GPU cards (GeForce RTX 3090 and 2070) were used to train the model, which took about two minutes per epoch for a M= 50,000 dataset.

## 3. Results and Discussion

In this section, we investigate how the RBM uses its weight matrix W and hidden layer h to encode the Boltzmann distributed states of the Ising model and what physical information can be extracted from machine learning concepts such as the visible energy and loss function.

### 3.1. Filters and Inverse Filters

It can be verified that the trained weight matrix elements $W_{ij}$ of a *T*-RBM follow a Gaussian distribution of zero mean with the largest variance at $T∼T_{c}$ (Figure 2a) [62]. The low temperature distribution here is different from the uniform distribution observed in [35], which results from the uniform initialization scheme used there. This suggests that the training of RBMs could converge to different minima when initialized differently. According to Equation (Equation 10), the biases $c_{i}$ and $b_{j}$ can be associated with the activation threshold of a hidden unit and a visible unit, respectively. For example, whether a hidden unit is activated ($h_{i}=+1$) or anti-activated ($h_{i}=-1$) depends on whether the incoming signal $w_{i}^{T}v$ from all visible units exceeds the threshold $-c_{i}$. The values of $c_{i}$ (and $b_{j}$) are all close to zero and are often negligible in comparison with the total incoming signal $w_{i}^{T}v$ (and $h^{T}w_{:,j}$) (see the Appendix A for the results of constrained RBMs where all biases are set to zero). The distribution of $c_{i}$ and $b_{j}$ should in principle be symmetric about zero (Figure 2b,c). A non-zero mean can be caused by an unbalanced dataset with an unequal number of m>0 and m<0 Ising configurations. The corresponding filter or inverse filter sum may also be distributed with a non-zero mean in order to compensate the asymmetric bias, as will be shown next.

Since v=s is an Ising configuration with ±1 units in our problem, $w_{i}^{T}v$ will be more positive (or negative) if the components of $w_{i}^{T}$ better match (or anti-match) the signs of the spin variables. In this sense, we can think of $w_{i}^{T}$ as a filter extracting certain patterns in Ising configurations. Knowing the representative spin configurations of the Ising model below, close to, and above the critical temperature $T_{c}$, we expect that $w_{i}^{T}$ ($i=1,2,\cdots ,n_{h}$) wrapped into an $L^{d}$ arrangement exhibits similar features. In Figure 3a, we show sample filters of *T*-RBMs with $n_{h}=400$ trained for the 2d Ising model at three temperatures T=1.0,2.25, and 3.5 (see the Appendix A for more examples of filters). At low *T*, the components of $w_{i}^{T}$ tend to be mostly positive (or negative), matching the spin up (or spin down) configurations in the ferromagnetic phase. At high *T*, filters $w_{i}^{T}$ possess strip domains consisting of roughly equal numbers of well-mixed positive and negative components, like Ising configurations during spinodal decomposition. Close to $T_{c}$, the $w_{i}^{T}$ patterns vary dramatically from each other, in accord with the large critical fluctuation. In particular, some even exhibit hierarchical clusters of various sizes. The element sum of the filter—*filter sum*
$\mathrm{sum}\left(w_{i}^{T}\right)=\sum_{j=1}^{n_{v}}W_{ij}$—plays a similar role as the magnetization *m*. The distribution of all the $n_{h}$ filter sums at each *T* changes with increasing temperature as the Ising magnetization changes, from bimodal to unimodal with the largest variance at $T_{c}$ (Figure 3b). This suggests that the peak of the variance $\langle {\left|\sum_{j=1}^{n_{v}}W_{ij}\right|}^{2}\rangle -{\langle \left|\sum_{j=1}^{n_{v}}W_{ij}\right|\rangle }^{2}$ as a function of temperature coincides with the Ising phase transition (inset of Figure 3b). More detailed results about the 2d and 3d Ising models are in the Appendix A.

When a hidden layer h is provided, the RBM reconstructs the visible layer v by applying the $n_{v}$ inverse filters $w_{:,j}$ ($j=1,2,\cdots ,n_{v}$) on h. The distribution of the *inverse filter sum*$\mathrm{sum}\left(w_{:,j}\right)=\sum_{i=1}^{n_{h}}W_{ij}$ is Gaussian with a mean close to zero (Figure 3c), where a large deviation from zero mean is accompanied by a non-zero average bias $\sum_{j}b_{j}/n_{v}$, as mentioned above (Figure 2b). We find that this is a result of the unbalanced dataset, which has ∼60% m<0 Ising configurations. Because the activation probability of a visible unit $v_{j}$ is determined by $w_{:,j}$, the correlation between visible units (Ising spins) is reflected in the correlation between inverse filters. This is equivalent to the analysis of the $n_{v}\times n_{v}$ matrix $W^{T}W$ or its eigenvectors as in [38,42], whose entries are the inner product $w_{:,j}^{T}w_{:,j^{\prime }}$ of inverse filters. We can therefore locate the Ising phase transition by identifying the temperature with the strongest correlation among the $w_{:,j}$s, e.g., the peak of $w_{:,j}^{T}w_{:,j^{\prime }}$ at a given distance $r_{jj^{\prime }}$ (inset of Figure 3c). See the Appendix A for results in 2d and 3d.

In contrast, the filters of the ∪T-RBM trained from 2d Ising configurations at all temperatures have background patterns like the high temperature *T*-RBM (in the paramagnetic phase). A clear difference is that most ∪T-RBM filters have one large domain of positive or negative elements (Figure 4a), similar to the receptive field in a deep neural network [29]. This domain randomly covers an area of the visual field of the L×L Ising configuration (see the Appendix A for all the $n_{h}$ filters). The existence of such domains in the filter causes the filter sum and the corresponding bias $c_{i}$ to be positive or negative with a bimodal distribution (Figure 4b,c). The inverse filter sum and its corresponding bias $b_{j}$ still have a Gaussian distribution, although the unbalanced dataset shifts the mean of $b_{j}$ away from zero.

### 3.2. Hidden Layer

Whether a hidden unit uses +1 or −1 to encode a pattern of the visible layer v is randomly assigned during training. In the former case, the filter $w_{i}^{T}$ matches the pattern ($w_{i}^{T}v$ is positive); in the latter case, the filter anti-matches the pattern ($w_{i}^{T}v$ is negative). For a visible layer v of magnetization *m*, the sign of $w_{i}^{T}v$ and the encoding $h_{i}$ is largely determined by the sign of $\mathrm{sum}(w_{i}^{T})$ (Table 1). Since the distribution of $\mathrm{sum}(w_{i}^{T})$ is symmetric about zero, the hidden layer of a *T*-RBM roughly consists of an equal number of +1 and −1 units—the “magnetization” $m_{h}=\frac{1}{n_{h}}\sum_{i=1}^{n_{h}}h_{i}$ of the hidden layer is always close to zero and its average $\langle m_{h}\rangle \approx 0$. The histogram of $m_{h}$ for all hidden encodings of visible states is expected to be symmetric about zero (Figure 5). We found that, for the smallest $n_{h}$, the histogram of $m_{h}$ at temperatures close to $T_{c}$ is bimodal due to the relatively large randomness of small hidden layers. As more hidden units are added, the two peaks merge into one and the distribution of $m_{h}$ becomes narrower. This suggests that a larger hidden layer tends to have a smaller deviation from $m_{h}=0$.

The order of the $h_{i}=\pm 1$ sequence in each hidden encoding h is arbitrary, but relatively fixed once the *T*-RBM is trained. The permutation of hidden units together with their corresponding filters (swap the rows of the matrix W) results in an equivalent *T*-RBM. Examples of hidden layers of *T*-RBMs with $n_{h}=400$ at different temperatures are shown in the inset of Figure 5, where the vector h is wrapped into a 20×20 arrangement. Note that there are actually no spatial relationships between different hidden units, and any apparent pattern in this 2d illustration is an artifact of the wrapping protocol.

As a generative model, a *T*-RBM can be used to produce more Boltzmann-distributed Ising configurations. Starting from a random hidden state $h^{\left(0\right)}$, this is often fulfilled by a sequence of Markov chain moves $h^{\left(0\right)}\to v^{\left(0\right)}\to h^{\left(1\right)}\to v^{\left(1\right)}\to \cdots$ until the steady state is achieved [31]. Based on the above-mentioned observations, we can design an algorithm to initialize $h^{\left(0\right)}$ that better captures the hidden encoding of visible states (equilibrium Ising configurations), thus enabling faster convergence of the Markov chain. After choosing a low temperature $T_{L}$ and a high temperature $T_{H}$, we generate the hidden layer as follows:At low $T\leq T_{L}<T_{c}$, if $\mathrm{sum}(w_{i}^{T})>0$, $h_{i}=+1$; if $\mathrm{sum}(w_{i}^{T})<0$, $h_{i}=-1$. This will be an encoding of an m>0 ferromagnetic configuration. To encode of an m<0 ferromagnetic configuration, just flip the sign of $h_{i}$.At high $T\geq T_{H}>T_{c}$, randomly assign $h_{i}=+1$ or −1 with equal probability. This will be an encoding of a paramagnetic configuration with m≈0.At intermediate $T_{L}<T<T_{H}$, to encode an m>0 Ising configuration, if $\mathrm{sum}(w_{i}^{T})>0$, assign $h_{i}=+1$ with probability $p_{h}\in \left(0.5,1.0\right)$ and $h_{i}=-1$ with probability $1-p_{h}$; if $\mathrm{sum}(w_{i}^{T})<0$, assign $h_{i}=-1$ with probability $p_{h}\in \left(0.5,1.0\right)$ and $h_{i}=+1$ with probability $1-p_{h}$. $p_{h}$ is a predetermined parameter, and the above two algorithms are just the special cases with $p_{h}=1.0$ ($T\leq T_{L}$) and $p_{h}=0.5$ ($T\geq T_{H}$), respectively. In practice, one may approximately use $p_{h}=(\langle |m|\rangle +1)/2$ or use linear interpolation within $T_{L}<T<T_{H}$, $p_{h}=0.5+0.5\left(T-T_{L}\right)/\left(T_{H}-T_{L}\right)$.

Below, we compare the (one-step) reconstructed thermal quantities using two different initial hidden encodings with results from a conventional multi-step Markov chain (Figure 6). The hidden encoding methods proposed here are quite reliable at low and high *T*, but less accurate at *T* close to $T_{c}$.

### 3.3. Visible Energy

When a *T*-RBM for temperature *T* is trained, we expect that $p_{\theta }\left(v\right)\approx p_{D}\left(v\right)\approx p_{T}\left(s\right)$—the Boltzmann distribution at that *T*. Although formally related to the physical energy in the Boltzmann factor (with temperature absorbed), the visible energy $E_{\theta }\left(v\right)$ of an RBM should be really considered as the negative log (relative) probability of a visible state v. For single-temperature *T*-RBMs, the mean visible energy $\langle E_{\theta }\left(v\right)\rangle$ increases monotonically with temperature (except for the largest $n_{h}$, which might be due to overfitting) (Figure 7a,b). The value of $\langle E_{\theta }\left(v\right)\rangle$ and its trend, however, cannot be used to identify the physical phase transition. In fact, $E_{\theta }\left(v\right)$ can differ from the reduced Hamiltonian $\frac{H(s)}{k_{B}T}$ by an arbitrary (temperature-dependent) constant while still maintaining the Boltzmann distribution $p_{\theta }\left(v\right)\approx p_{T}\left(s\right)$ (if the partition function $Z_{\theta }$ is calibrated accordingly).

The trend of $\langle E_{\theta }\left(v\right)\rangle$ for *T*-RBMs can be understood by considering following approximate forms. First, due to the symmetry of +1 and −1, the biases $b_{j}$ and $c_{i}$ are all close to zero. A constrained *T*-RBM with zero bias has a visible energy:(17)$$\begin{array}{c} E_{W}\left(v\right)=-\sum_{i=1}^{n_{h}}\ln\left(e^{-w_{i}^{T}v}+e^{w_{i}^{T}v}\right) \end{array}$$
that approximates the visible energy of the full *T*-RBM, i.e., $E_{\theta }\left(v\right)\approx E_{W}\left(v\right)$. Next, unless $w_{i}^{T}v$ is close to zero, one of the two exponential terms in Equation (Equation 17) always dominates such that $E_{W}\left(v\right)\approx {\tilde{E}}_{W}\left(v\right)$, where
(18)$$\begin{array}{c} {\tilde{E}}_{W}\left(v\right)=-\sum_{i=1}^{n_{h}}\left|w_{i}^{T}v\right|=-\sum_{i=1}^{n_{h}}\left|\sum_{j=1}^{n_{v}}W_{ij}v_{j}\right|. \end{array}$$

Equation (Equation 18) can further be approximated by setting v=1 with all $v_{j}=+1$, i.e., ${\tilde{E}}_{W}\left(v\right)\approx {\tilde{E}}_{W}\left(1\right)$ with
(19)$$\begin{array}{c} {\tilde{E}}_{W}\left(1\right)=-\sum_{i=1}^{n_{h}}\left|\mathrm{sum}(w_{i}^{T})\right|=-\sum_{i=1}^{n_{h}}\left|\sum_{j=1}^{n_{v}}W_{ij}\right|. \end{array}$$

In summary, $E_{W}\left(v\right)$, ${\tilde{E}}_{W}\left(v\right)$, and ${\tilde{E}}_{W}\left(1\right)$ are all good approximations to the original $E_{\theta }\left(v\right)$ (Figure 7a). The increase of mean $\langle E_{\theta }\left(v\right)\rangle$ with temperature coincides with the increase of $-\left|\mathrm{sum}(w_{i}^{T})\right|$ with temperature, which is evident from Figure 3b. At fixed temperature, the decrease of $\langle E_{\theta }\left(v\right)\rangle$ with $n_{h}$ is a consequence of the sum $\sum_{i=1}^{n_{h}}$ in the definition of visible energy. The variance $\langle E_{\theta }^{2}\rangle -{\langle E_{\theta }\rangle }^{2}$ is a useful quantity for phase transition detection, because it reflects the fluctuation of the probability $p_{\theta }\left(v\right)$. In both low *T* ferromagnetic and high *T* paramagnetic regimes, $p_{\theta }\left(v\right)$ is relatively homogeneous among different states. When *T* is close to $T_{c}$, the variance of $p_{\theta }\left(v\right)$ and $E_{\theta }\left(v\right)$ is expected to peak (Figure 7d,e). The abnormal rounded (and even shifted) peaks at large $n_{h}$ could be a sign of overfitting.

For the all-temperature ∪T-RBM, the Ising phase transition can be revealed by either the sharp increase of the mean $\langle E_{\theta }\left(v\right)\rangle$ or the peak of the variance $\langle E_{\theta }^{2}\rangle -{\langle E_{\theta }\rangle }^{2}$ (Figure 7c,f). However, this apparent detection can be a trivial consequence of the special composition of the dataset $D_{\cup T}$, which contains Ising configurations at different temperatures in equal proportion. Only configurations at a specific *T* are fed into the model to calculate the average quantity at that *T*. Technically, a visible state v in $D_{\cup T}$ is not subject to the Boltzmann distribution at any specific temperature. Instead, the true ensemble of $D_{\cup T}$ is a collection of $n_{T}$ different Boltzmann-distributed subsets. Many replicas of the same or similar ferromagnetic states are in $D_{\cup T}$, giving rise to a large multiplicity, high probability, and low visible energy for such states. In comparison, high temperature paramagnetic states are all different from each other and, therefore, have low $p_{\theta }\left(v\right)$ (high $E_{\theta }\left(v\right)$) for each one of them. Knowing this caveat, one should be cautious when monitoring the visible energy of a ∪T-RBM to detect phase transition, because changing the proportion of Ising configurations at different temperatures in $D_{\cup T}$ can modify the relative probability of each state.

### 3.4. Pseudo-Likelihood and Entropy Estimation

The likelihood L(θ) defined in Equation (Equation 11) is conceptually equivalent to the physical entropy *S* defined by the Gibbs entropy formula, apart from the Boltzmann constant $k_{B}$ difference (Appendix B). However, just as entropy *S* cannot be directly sampled, the exact value of L(θ) is not accessible. In order to estimate *S*, we calculated the pseudo-likelihood $\tilde{L}\left(\theta \right)$ instead, which is based on the mean-field-like approximation $p_{\theta }\left(v\right)\approx \prod_{i=1}^{n_{v}}p_{\theta }\left(v_{i}|v_{j\neq i}\right)$. Similar ideas to estimate the free energy or entropy were put forward with the aid of variational autoregressive networks [63] or neural importance sampling [64]. The true and estimated entropy of the 2d and 3d Ising models using *T*-RBMs with different $n_{h}$ are shown in Figure 8a,b. As a comparison, we also considered a “pseudo-entropy” with a similar approximation:(20)$$\tilde{S}=-k_{B}{\langle \sum_{i=1}^{N}p_{T}\left(s_{i}|s_{j\neq i}\right)\rangle }_{s∼p_{T}}\approx S$$
where the conditional probability:(21)$$p_{T}\left(s_{i}|s_{j\neq i}\right)=\frac{e^{-\frac{H(s)}{k_{B}T}}}{e^{-\frac{H(s)}{k_{B}T}}+e^{-\frac{H(\left[s_{1},\cdots ,-s_{i},\cdots ,s_{N}\right])}{k_{B}T}}}$$
and the ensemble average ${\langle \cdots \rangle }_{s∼p_{T}}$ is taken over states obtained from Monte Carlo sampling. In both 2d and 3d, $\tilde{S}$ is lower than the true *S*, especially at high *T*, because a mean-field treatment tends to underestimate fluctuations.

While increasing model complexity by adding hidden units is usually believed to reduce the reconstruction error, e.g., of energy and heat capacity [35,36] (see also the Appendix A), a recent study suggested that a trade-off could exist between the accuracy of different statistical quantities [65]. Here, we found that the pseudo-likelihood of *T*-RBMs with the fewest hidden units in our trials ($n_{h}=400$) appears to provide the best prediction for entropy. Increasing $n_{h}$ leads to larger deviations from the true *S* at higher *T*. The decreasing of $\tilde{L}$ with $n_{h}$ at fixed temperature agrees with the trend of the visible energy. A lower $E_{\theta }\left(v\right)$ corresponds to a higher $p_{\theta }\left(v\right)$ and, thus, a lower $\tilde{L}$ according to its definition. The surprisingly good performance of $\tilde{L}$ in approximating *S* could be due to the fact that visible units $v_{i}$ in RBMs are only indirectly correlated through hidden units, which collectively serve as an effective mean-field on each visible unit. We also calculated $\tilde{L}\left(\theta \right)$ with the all-temperature ∪T-RBM in 2d (Figure 8c). Compared with single-temperature *T*-RBMs of the same $n_{h}$ (Figure 8a), the ∪T-RBM predicts higher $\tilde{L}\left(\theta \right)$ with considerable deviations even at low *T*. The trend of $\tilde{L}\left(\theta \right)$ also agrees with that of $\langle E_{\theta }\left(v\right)\rangle$ (Figure 7c).

The knowledge of the entropy allows us to estimate the phase transition point according to the thermodynamic relation $C_{V}=T\frac{dS}{dT}$. We constructed this estimated $C_{V}$ as a function of temperature using $\tilde{L}\left(\theta \right)$ and its numerical fitting, whose peaks are expected to be located at $T_{c}$ (Figure 9). The predicted $T_{c}$s are compared with the results from entropy and pseudo-entropy, as well as the Monte Carlo simulation results for our finite systems and the known exact values for infinite systems in Table 2. It can be seen that single-temperature *T*-RBMs capture the transition point fairly well within an error of about 1–3%.

## 4. Conclusions

In this work, we trained RBMs using equilibrium Ising configurations in 2d and 3d collected from Monte Carlo simulations at various temperatures. For single-temperature *T*-RBMs, the filters (row vectors) and the inverse filters (column vectors) of the weight matrix exhibit different characteristic patterns and correlations, respectively, below, around, and above the phase transition. These metrics, such as filter sum fluctuation and inverse filter correlation, can be used to locate the phase transition point. The hidden layer h on average contains an equal number of +1 and −1 units, whose variance decreases as more hidden units are added. The sign of a particular hidden unit $h_{i}$ is determined by the signs of the filter sum $\mathrm{sum}(w_{i}^{T})$ and the magnetization *m* of the visible pattern. However, there is no spatial pattern in the sequence of positive and negative units in a hidden encoding.

The visible energy reflects the relative probability of visible states in the Boltzmann distribution. Although the mean of visible energy is not directly related to the (physical) internal energy and does not reveal a clear transition, its fluctuation, which peaks at the critical point, can be used to identify the phase transition. The value and trend of the visible energy can be understood from its several approximation forms, in particular the sum of the absolute value of filter sums. The pseudo-likelihood of RBMs is conceptually related to and can be used to estimate the physical entropy. Numerical differentiation of the pseudo-likelihood provides another estimator of the transition temperature because it provides an estimate of the heat capacity. All these predictions about the critical temperature were made by unsupervised RBM learning, for which human labeling of the phase types is not needed.

As a comparison, we also trained an all-temperature ∪T-RBM, whose dataset is a mixture of Boltzmann-distributed states over a range of temperatures. Each filter of this ∪T-RBM is featured by one large domain in its receptive field. Although the visible energy and pseudo-likelihood of the ∪T-RBM show a certain signature of the phase transition, one should be cautious, as this detection could be an artifact of the composition of the dataset. Changing the proportions of Ising configurations at different temperatures could bias the probability and the transition learned by the ∪T-RBM.

By extracting the underlying (Boltzmann) distribution of the input data, RBMs capture the rapid (phase) transition of such a distribution as the tuning parameter (temperature) is changed, without knowledge of the physical Hamiltonian. Information about the distribution is completely embedded in the configurations and their frequencies in the dataset. It would be interesting to see if such a general scheme of RBM learning can be extended to study other physical models of phase transition.

## Figures and Tables

**Figure 1 entropy-24-01701-f001:**
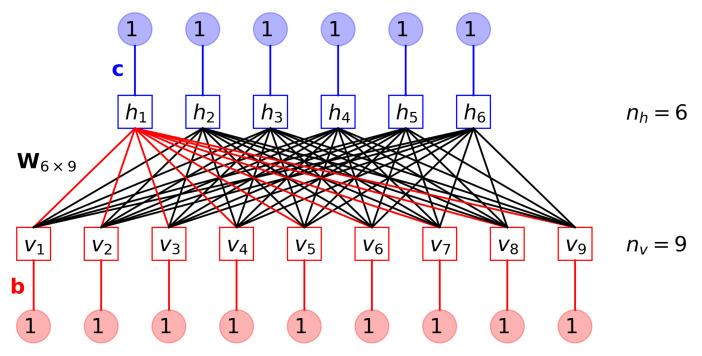
A restricted Boltzmann machine (RBM) with $n_{h}=6$ hidden units and $n_{v}=9$ visible units. Model parameters θ={W,b,c} are represented by connections. A filter $w_{1}^{T}$ from visible units to the first hidden unit is highlighted by red (light color) connections.

**Figure 2 entropy-24-01701-f002:**
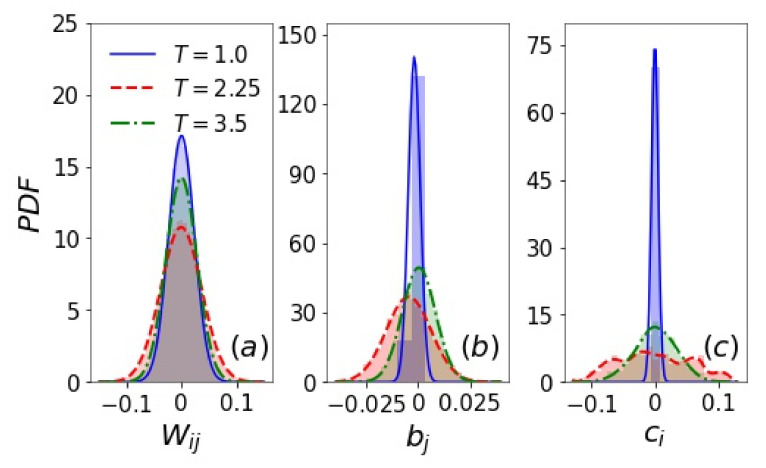
Probability density function (PDF) of the distribution of (**a**) $W_{ij}$, (**b**) $b_{j}$, and (**c**) $c_{i}$ of *T*-RBMs with $n_{h}=400$ hidden units for the 2d Ising model at temperatures below, close to, and above $T_{c}$.

**Figure 3 entropy-24-01701-f003:**
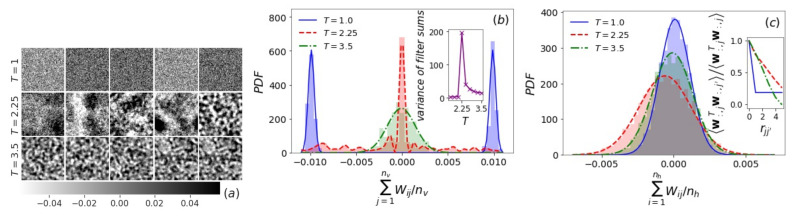
*T*-RBMs with $n_{h}=400$ for the 2d Ising model at temperature T=1.0, 2.25, and 3.5. (**a**) Five sample filters $w_{i}^{T}$ at each temperature. The color bar range is set to be within about two standard deviations of the distribution. (**b**) PDF of the distribution of the $n_{h}=400$ filter sums (normalized by $n_{v}$). Inset: variance $\langle {\left|\sum_{j=1}^{n_{v}}W_{ij}\right|}^{2}\rangle -{\langle \left|\sum_{j=1}^{n_{v}}W_{ij}\right|\rangle }^{2}$ of the filter sum as a function of temperature. (**c**) PDF of the distribution of the $n_{v}=4096$ inverse filter sums (normalized by $n_{h}$). Inset: correlation between a pair of inverse filters $w_{:,j}$ and $w_{:,j^{\prime }}$ (normalized by auto-correlation) as a function of spin–spin distance $r_{jj^{\prime }}$.

**Figure 4 entropy-24-01701-f004:**
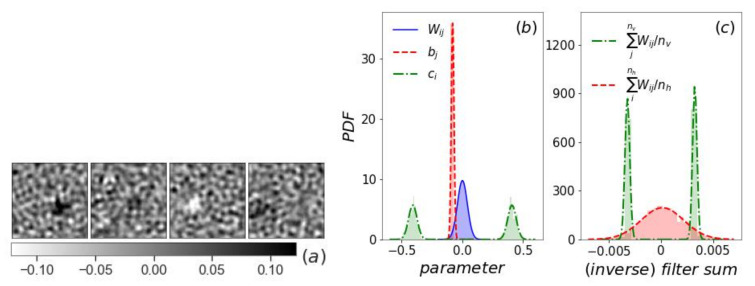
The ∪T-RBM with $n_{h}=400$ for the 2d Ising model. (**a**) Four sample filters $w_{i}^{T}$. (**b**) PDF of the distribution of $W_{ij}$, $b_{j}$, and $c_{i}$. (**c**) PDF of the distribution of the $n_{h}=400$ filter sums and the $n_{v}=4096$ inverse filter sums.

**Figure 5 entropy-24-01701-f005:**
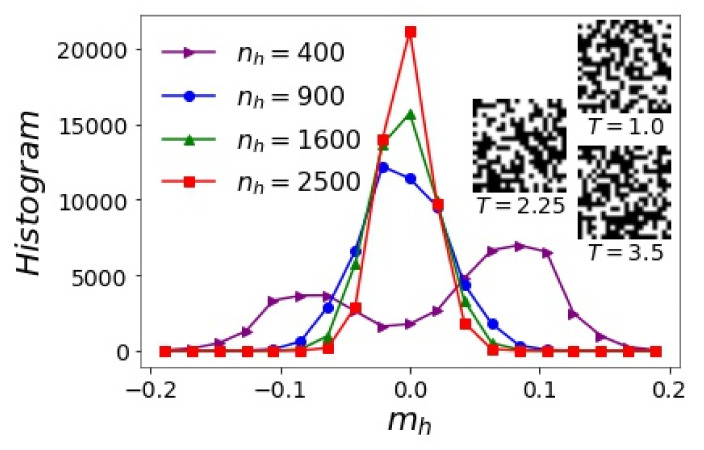
Histogram of $m_{h}$ obtained from the hidden encodings of M= 50,000 2d Ising configurations at T=2.25 using *T*-RBMs with various $n_{h}$. Inset: examples of the hidden layer of *T*-RBMs with $n_{h}=400$ wrapped into a 20×20 matrix at three temperatures, where +1/−1 units are represented by black/white pixels.

**Figure 6 entropy-24-01701-f006:**
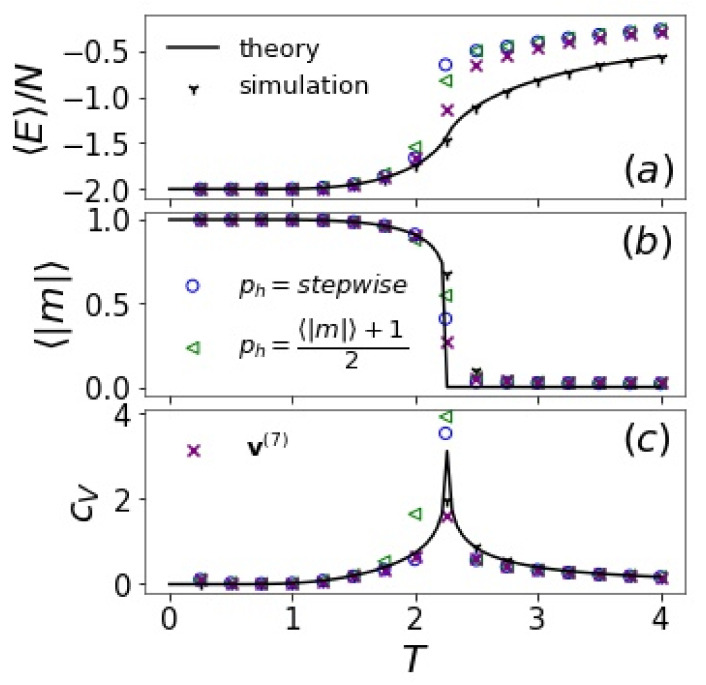
(**a**) Internal energy, (**b**) magnetization, and (**c**) specific heat of 2d Ising states reconstructed by *T*-RBMs ($n_{h}=400$) with the hidden layer $h^{\left(0\right)}$ initiated according to $p_{h}=(\langle |m|\rangle +1)/2$ or $p_{h}=1.0\left(T\leq 2.0\right),0.5\left(T\geq 2.5\right),0.75\left(2.0<T<2.5\right)$ (stepwise). Reconstruction by a seven-step Markov chain from random $h^{\left(0\right)}$ is compared ($v^{\left(7\right)}$). Analytical and Monte Carlo simulation results are also shown.

**Figure 7 entropy-24-01701-f007:**
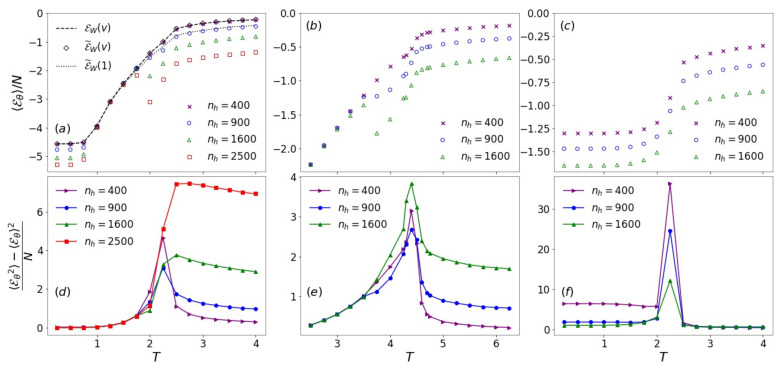
Mean and variance of visible energy $E_{\theta }$ as a function of temperature for 2d (**a**,**c**,**d**,**f**) and 3d (**b**,**e**) Ising models captured by *T*-RBMs (**a**,**b**,**d**,**e**) and the ∪T-RBM (**c**,**f**) of various hidden neurons $n_{h}$. Three approximate forms of visible energy for $n_{h}=400$*T*-RBMs are shown in (**a**).

**Figure 8 entropy-24-01701-f008:**
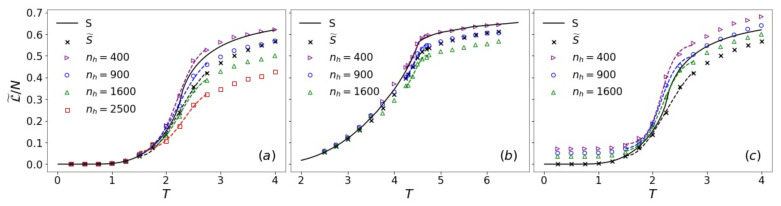
Pseudo-likelihood $\tilde{L}$ per spin of *T*-RBMs (**a**,**b**) and of the ∪T-RBM (**c**) with different numbers $n_{h}$ of hidden units for the 2d (**a**,**c**) and 3d (**b**) Ising models in comparison with entropy *S* and pseudo-entropy $\tilde{S}$ per spin. Dashed lines are polynomial fittings around $T_{c}$.

**Figure 9 entropy-24-01701-f009:**
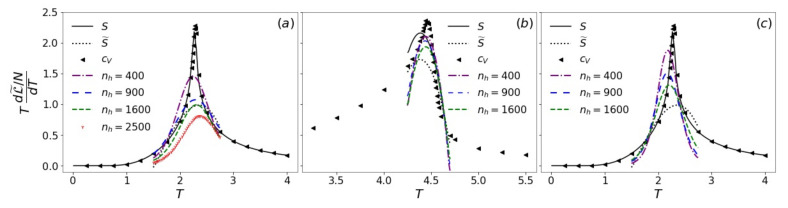
$T\frac{d\tilde{L}}{dT}$ per spin of *T*-RBMs (**a**,**b**) and of the ∪T-RBM (**c**) with different numbers $n_{h}$ of hidden units for the 2d (**a**,**c**) and 3d (**b**) Ising models in comparison with $T\frac{dS}{dT}$ and $T\frac{d\tilde{S}}{dT}$ per spin, as well as specific heat $c_{V}$ calculated from Monte Carlo simulation.

**Table 1 entropy-24-01701-t001:** When $\mathrm{sum}(w_{i}^{T})>0$, a visible layer pattern v with magnetization m>0 (or m<0) is more likely to be encoded by a hidden unit $h_{i}=+1$ (or $h_{i}=-1$). When $\mathrm{sum}(w_{i}^{T})>0$, the encoding is opposite.

	$\mathrm{sum}(w_{i}^{T})>0$	$\mathrm{sum}(w_{i}^{T})<0$
m>0	$h_{i}=+1$	$h_{i}=-1$
m<0	$h_{i}=-1$	$h_{i}=+1$

**Table 2 entropy-24-01701-t002:** $T_{c}$ estimated according to the peak of $T\frac{d\tilde{L}}{dT}$ obtained from single-temperature *T*-RBMs and the all-temperature ∪T-RBM with different numbers ($n_{h}$) of hidden units. Predictions from numerical derivatives $T\frac{dS}{dT}$ and $T\frac{d\tilde{S}}{dT}$ are also shown for comparison. Results extracted from the peak of $c_{V}$ obtained by Monte Carlo simulations of finite systems are listed under “MC”. “Exact” refers to analytical or numerical results for infinite systems.

Model	$n_{h}=400$	900	1600	2500	*S*	$\tilde{S}$	MC	Exact
2d*T*-RBM	2.240	2.291	2.316	2.367	2.267	2.367	2.28	2.269
2d∪T-RBM	2.189	2.163	2.214	-	2.267	2.367	2.28	2.269
3d*T*-RBM	4.444	4.434	4.444	-	4.390	4.383	4.44	4.511

## Data Availability

Not applicable.

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
