# Peer review of "Thermodynamics of the Ising Model Encoded in Restricted Boltzmann Machines"

_entropy, 2022, doi:10.3390/e24121701_

Round 1

Reviewer 1 Report

The manuscript addresses the problem of learning the thermodynamics of an Ising model using a restricted Boltzmann machine.

The analysis is standard, the results are expected and quite unsurprising: the problem itself is not new nor particularly original.

I do not recommend this manuscript for publication in light of a significant lack of originality

Reviewer 2 Report

The manuscript presents an application of restricted Boltzmann machines (RBM) to equilibrium Ising model trying in order to verify if it is possible to use the machine to generate spin configurations with the same statistical properties of the Ising model namely reproducing the average value of observed physical quantities and thus being able to find the occurrence of a phase transition. The construction of RBMs to learn the Ising model probability distribution for specific temperatures has been previously reported by other authors (Ref 35). In the present work much larger system sizes are considered (L=64 for two dimensions and L=16 for three dimensions, corresponding to N=4096 spin variables. Ref 35 considered N=64 spin variables). The authors define the additional goal to understand how the various components of the RBM contribute to the learning of the Ising model probability distribution. The authors present a comprehensive description of the RBM and the methods used to adjust the parameters of the machine. They build a RBM for each temperature feeding the learning process with a dataset generated by a Monte Carlo simulation of the Ising model at a specific temperature. They considered as well a dataset with all temperature spin configurations to build an all temperature RBM.

I have some coments I would like to see answered.

1) Since no details are presented of the generation of the dataset I would like to know what precautions were taken for generating the M=50000 spin configurations in a way that the correlations between the configurations arisen from the Markov chain Monte Carlo are negligible and how such correlations could affect the RBMs constructed.

2) The authors look at the histogram of weight values Wij obtained at a given temperature, and they say (lines 132-134 of manuscript) that they obtain a high temperature histogram different than the one obtained in Ref 35. Please clarify what differences you found and what explanation for the difference can be given. The high temperature distribution shown in Fig. 3 of Ref 35 is peaked near zero and not uniform as written in line 133. Zero weighting matrix values are expected for high temperature independent spin variables.

3) The prediction of average Ising model energy as a function of temperature obtained from the RBM acting as a generative model seem not to be very accurate specially at high temperatures. The authors consider a single step Markov chain generation of the visible states starting from a specific initialized hidden state (with two initialization methods) and a seven step Markov chain starting with a random hidden variable initialization. What happens if you increase the number of Markov chain steps in each case? Do the results improve? Why you have not studied the susceptibility?

4) Please explain how the average visible energy is calculated. Is it from the generated RBM visible configurations or do you use the specific temperature dataset configurations to make the average? Would the results be different when averaged in these two different ways? In the line 255 you say that for the all-T RBM machine only configurations with a specific T are fed into the model to obtain results for a specific T. is this the only way the all-T RBM machine can produce T dependent results.

5) Please explain how you can use the all T - RBM machine to predict average ising energy, magnetization and specific heat as you present in the Fig. 4 of supplementary material.

6) In table 2 the exact results for two dimensions you compare with are for infinite systems. The 3d result for Tc is a precise infinite system extrapolation obtained from simulation but not exact. Since you present both d=2 and d=3 simulation results to compare with you see the effects of the finite size. However in table 2 the comparison with infinite system Tc may be somewhat misleading because (even exact) predictions for Tc of N=4096 systems are different.

In summary, I consider that the results presented in the manuscript are interesting, that the manuscript is well written and that the results are technically sound. I recommend the manuscript for publication but I would like the points raised above to be clarified.

Reviewer 3 Report

The authors studied the internal structure of RBM trained with the Ising spin configurations across the phase transition in the 2D and 3D Ising models. The authors found that the internal machine parameters of RBM have different patterns across the transition that can be used to determine the transition point. The paper is written well, and its analysis of the internal structure of RBM is new and thorough, which I believe can help advance the knowledge of how machine learning works with the data across the phase transition. On these grounds, I recommend the publication of this paper in Entropy.

Although, there are some minor comments that the authors may consider before publication.

(1) I believe that the authors want to present the meaningfulness of their study by finding the critical point of 2D and 3D Ising models within their machine learning analysis. Table 2 certainly has that information, but the readers may want to see how those numbers are obtained more directly in the main text. Moving the figure of T_c estimation currently in Chapter V of the supplementary material into the main paper with some more discussions would help increase the readability. 

(2) No error bars are provided in this paper's figure. Since it is based on the Monte Carlo data, and the machine learning procedures have a stochastic nature, adding error bars would increase the readability and concreteness of the messages the authors want to deliver in this paper.
